# A Generative Adversarial and Spatiotemporal Differential Fusion Method in Radar Echo Extrapolation

Xianghua Niu [1], Lixia Zhang [2,*], Chunlin Wang [3,4], Kailing Shen [3,4], Wei Tian [3,4] and Bin Liao [1]

1  State Key Laboratory of Geo-Information Engineering, Xi'an 710054, China; niuxiangh@gmail.com (X.N.); liaobin198@126.com (B.L.)
2  Shijiazhuang Meteorological Bureau, Shijiazhuang 050081, China
3  School of Computer and Software, Nanjing University of Information Science and Technology, Nanjing 210044, China; 20211221036@nuist.edu.cn (C.W.); skling@nuist.edu.cn (K.S.); tw@nuist.edu.cn (W.T.)
4  Engineering Research Center of Digital Forensics, Ministry of Education, Nanjing University of Information Science and Technology, Nanjing 210044, China
*  Correspondence: 13933119682@163.com

**Abstract:** As an important part of remote sensing data, weather radar plays an important role in convective weather forecasts to reduce extreme precipitation disasters. The existing radar echo extrapolation methods do not utilize the local natural characteristics of the radar echo effectively but only roughly extract the whole characteristics of the radar echo. To address these challenges, we design a spatiotemporal difference and generative adversarial fusion model (STDGAN). Specifically, a spatiotemporal difference module (STD) is designed to extract local weather patterns and model them in detail. In our model, spatiotemporal difference information and spatiotemporal features captured by the model itself are fused together. In addition, our model is trained in a generative adversarial network (GAN) framework; it helps to generate a clearer map of future radar echoes at the image level. The discriminator consists of multi-scale feature extractors, which can simulate weather models of various scales more completely. Finally, extrapolation experiments were conducted using actual radar echo data from Shijiazhuang and Nanjing. The experiments have shown that our model has a more accurate prediction performance for predicting local weather patterns and overall echo change trajectories compared with previous research models. Our model achieved MSE, PSNE, and SSIM values of 132.22, 37.87, and 0.796, respectively, on the Shijiazhuang radar echo dataset. In addition, our model also showed better performance results on the Nanjing radar echo dataset. The results show that the MSE was 49.570, the PSNR was 0.714, and the SSIM was 30.633. The CC value was 0.855.

**Keywords:** radar extrapolation; generative adversarial network; difference; spatiotemporal fusion

## 1. Introduction

Extreme precipitation is one of the important factors causing natural disasters, which has a profound impact on every aspect of people's life. Accurate and timely forecasts of the coming extreme precipitation can avoid economic losses and help protect people's lives and ensure the safety of property [1–6]. The precipitation forecast is a prediction of rainfall (and other precipitation phenomena) in the next 1–2 h. It can provide timely and accurate information for weather-related decisions in various departments. At the same time, it can also improve public safety and reduce economic losses caused by extreme weather events such as floods, landslides, and hail. Remote sensing data are an important data source for observing meteorological phenomena. Among them, weather radar can effectively observe precipitation. The precipitation retrieved by a radar echo series can forecast the precipitation in the next 1–2 h. It can provide information about the development and change in precipitation and help to correctly judge the possible impact of precipitation [7]. However, the spatiotemporal characteristics of the precipitation development process have

great uncertainty because it involves the complex and nonlinear spatiotemporal dynamics of precipitation phenomena [8]. This makes it difficult to model and predict accurately. Traditional methods based on radar echo extrapolation include TREC, SCIT, and the optical flow method [9–11]. TREC focuses on the prediction of the motion vector of the radar echo mode, and it is widely used in ITWS, NIMROD, NCAR ANC, and other near forecast systems. However, TREC is susceptible to cluttering and small-scale changes in the radar echoes, resulting in distorted radar echo images. The SCIT algorithm can identify a strong storm and predict the position of the next storm by linear extrapolation based on the position of the center of mass of the storm in the past time. However, when radar echoes fuse and split, the prediction accuracy decreases rapidly. Optical flow methods, such as ROVER, calculate the optical flow field from the continuous time radar echo image. It replaces the radar echo motion vector field with the optical flow field and pushes the radar echo off-site based on the motion vector to achieve the proximity prediction. However, these methods only infer the echo position of the next moment from the radar echo images of the previous few moments; they ignore the motion nonlinearity of the small- and medium-scale convective system in the actual radar echo. In recent years, deep learning methods have been gradually applied to various fields, including the precipitation forecast business. Compared with traditional radar echo extrapolation methods, methods based on deep learning can independently learn the spatiotemporal features from the radar echo data without relying on the features of physical assumptions.

We summarize the main causes of the forecasting dilemma as follows. First of all, the changes of the atmospheric system are complex and diverse, full of uncertainty and chaos. This is more challenging than the normal spatiotemporal series prediction task. Moreover, because the results of the forecast will be used as a basis for the prediction of the more distant future, this inevitably leads to large systematic errors that are constantly iterated. Finally, as one of the motivations for this study, we believe that previous studies have neglected local weather types with great forecasting value. This seriously affects the trend prediction of overall weather conditions.

In this study, we propose a new spatiotemporal difference model based on a generative adversarial network (STDGAN) for radar echo extrapolation. First, in the framework of a generative adversarial network, a generator network is responsible for using the historical radar echo sequence as an input to predict the future radar echo pattern, and a discriminator network is responsible for distinguishing the predicted data from the real future data. In addition, for the generator network, we integrate spatiotemporal difference signals into the structure of the cyclic unit (the basic unit of the RNN model) to strengthen the spatiotemporal feature extraction and extrapolation ability of the model. For the discriminator network, we use the multi-spatial scale feature extraction method to effectively identify the large-scale and small-scale radar echo features and output category information. Finally, the distance between the real data distribution and the predicted data distribution is constrained by the loss function to generate a clearer and more detailed radar echo map. In order to evaluate our network, we conducted extrapolation experiments on the actual radar echo data provided by the Shijiazhuang Meteorological Bureau and Nanjing University, which are more in line with the actual business needs.

The results show that our model can generate more accurate and detailed prediction results than the previous models of radar echo extrapolation tasks, and it achieve the best results.

## 2. Related Work

In recent years, deep learning technology has been widely applied to analyze, learn, and reason uncertain problems in various fields [12–17]. Convective weather forecasting is a series of forecasting problems based on time and space. Some scholars have applied deep learning technology to weather forecasts and achieved satisfactory results. Traditional algorithms such as the optical flow method have the advantages of a good real-time performance and support from physical theory in radar echo extrapolation tasks [18,19].

However, traditional algorithms are not general enough for radar echo extrapolation tasks and cannot achieve automatic feature extraction. In contrast, deep learning technology can make up for the shortcomings of traditional algorithms to a certain extent and can adapt to complex and changeable environments. Spatiotemporal prediction based on deep learning involves two important aspects: spatial correlation and temporal dynamics. The performance of a prediction system depends on its memory and on the reasoning ability of the relevant structural information.

The convolutional neural network method converts the input image sequence into one or more frame image sequences on a certain channel [20]. Many scholars have proposed implementation schemes based on this method. Kalchbrenner et al. proposed a probabilistic video model called video pixel network (VPN) [21]. Xu et al. proposed a PredCNN network that stacked multiple extended causal convolution layers [22]. Ayzel et al. proposed a CNN model named DozhdyaNet [23]. Some works have introduced UNet [24] for radar echo prediction. It was originally proposed in the field of medical image segmentation. Specifically, RainNet [13] modified the network structure of the last layer of UNet to adapt to pixel level radar echo prediction tasks. FURENet added two additional encoders to UNet for multimodal learning [2]. In addition, SmaAt UNet [24] added attention modules and deep separable convolutions on UNet [24]. Compared with the traditional radar echo extrapolation method, the method based on convolutional neural networks can make use of a large number of historical radar echo observation data to learn its spatial variation law, including the strengthening and weakening process of rainfall intensity. The convolutional neural network is not sensitive to the change in the time dimension, and the prediction mode is relatively fixed. Therefore, the method based on neural networks has some limitations and is not widely used in radar echo extrapolation.

At present, the neural network models used for radar echo extrapolation are mainly image sequence prediction methods based on recurrent neural networks (RNNs). Shi et al. proposed ConvLSTM to replace the Hadamard multiplier by using convolution operations in LSTM internal transformations [1]. This extends the time series prediction task to the spatiotemporal series prediction task and further extracts the spatiotemporal feature information. Subsequently, many variations of ConvLSTM were proposed. TrajGRU introduced the idea of the optical flow method to dynamically capture the movement trend of radar echoes; however, this method would consume a lot of computing resources and training speed [25]. PredRNN divides the originally unified memory state into spatial memory and temporal memory [26]. Memory states in both directions participate in transforming recursive units at the same time, thus further enhancing the ability of spatial information to propagate in the spatiotemporal dimension. In addition, information is transmitted horizontally and vertically through highway connections, which helps to model spatiotemporal dynamics. In order to coordinate the learning of long and short frames, Wang et al. further proposed PredRNN++ to increase the depth of recursive units and improve the modeling ability of the model for spatiotemporal information in a cascading manner [27]. In order to enhance the model's ability to model higher-order dynamics, MIM introduced differential signals to process nonstationary and stationary information [28]. MotionRNN inserts a MotionGRU between stacked PredRNN layers, which assumes that the physical world motion can be decomposed into transient changes and motion trends [29]. Jing et al. designed a hierarchical prediction RNN model for the long-term extrapolation of radar echoes, which adopted a hierarchical prediction strategy and a coarse-to-fine cycle mechanism to reduce the prediction errors accumulated over time in the long-term extrapolation [3]. However, these models are all universal spatiotemporal series prediction models that do not take into account the inherent uncertainty and prediction ambiguity caused by uncertainty in radar echo sequences. In addition, previous studies have ignored the importance of small-scale weather patterns.

In recent years, generative adversarial networks [30] (GANs) have achieved great success in the field of image generation, which has greatly improved the quality of generated images. GAN is mainly divided into generators and discriminators. The generator

is responsible for generating images and attempting to deceive the discriminator. The discriminator attempts to identify the generated image and from real images. These two systems enhance the credibility of the generated images by playing games with each other. However, if two training systems are not synchronized properly, the generator and discriminator may lose the gradient to be updated. To address this drawback, a more robust version of Wasserstein GAN (WGAN) has been developed [31]. Due to the excellent performance of GAN in image generation, several recent studies have also utilized GAN as regularization in the loss function to generate more qualified predictions for precipitation forecasting. These methods roughly first use convolutional neural network models for prediction, and then establish GAN methods to further correct the prediction to improve its clarity [32,33]. Although these existing studies have demonstrated the effectiveness of GAN, their powerful potential has not been fully utilized due to the lack of effective design of generators and discriminators in network structure.

## 3. Methods

Our proposed model focuses on solving the problem of ambiguity in unsupervised precipitation proximity forecasting and the dilemma of previous models not being able to accurately perceive meteorological changes (such as the movement trend of cloud clusters) and make accurate predictions.

### 3.1. Problem Description

Considering that radar has the advantages of high spatial and temporal resolution, wide geographical coverage, and real-time data transmission, rainfall prediction mainly relies on the radar echo as the observed value. Then, in an indirect way, the reflectivity of the radar echo is translated into predicted rainfall. Therefore, the task of rainfall prediction can be regarded as utilizing the observation value of the historical radar echo to predict the evolution trend of the future radar echo. In this study, the reflectivity factor intensity of the radar echo is reflected by the pixel value in the form of grid points.

In the rain forecast, we observe that the radar echo movement is in a discrete state. Therefore, we consider a sample of radar echoes observed at time intervals:

$$S = (S_{t0}, S_{(t0+\Delta t)}, \ldots, S_{t1}), S \in R^{C \times H \times W}, t_1 = t_0 + v\Delta t \tag{1}$$

The real future sequence is $S_{\text{target}} = (S_{t_{1+1}}, S_{t_{1+2}}, \ldots, S_{t_{1+n}})$, the forecasted future sequence of $S_{pred} = (\hat{S}_{t_{1+1}}, \hat{S}_{t_{1+2}}, \ldots, \hat{S}_{t_{1+n}})$, where $n$ is the predicted number of frames. In this study, we try to make the predicted future radar echo sequence approximate the true value by using a neural network with $\theta$ as the parameter. The formulaic description of the problem we are studying is

$$\theta = \arg\max_{\theta} P(S_{target}|S; \theta) \tag{2}$$

### 3.2. Generative Adversarial Networks

In previous studies, the generative adversarial network (GAN) is often used as a generative model [30]. The generative adversarial network mainly consists of generator network G and discriminator network D. The generator network G takes random vector Z satisfying the normal distribution as the input and generates reconstructed false data G(Z) through complex nonlinear changes. The discriminator network D receives the real data and the data generated by the generator network G as the input and outputs the category information (0 or 1), respectively. During training, the model can generate more realistic data through the confrontation and game between the two networks. The adversarial training is defined as the following min-max optimization problem.

$$G^*, D^* = \arg\min_{g}\max_{c} V(G, D) \tag{3}$$

The loss function of the GAN network is

$$\min_{G} \max_{D} V(D, G) = E_{x \sim p_{true}}[logD(x)] + E_{z \sim p(z|\mu;\sigma)}[\log(1 - D(G(z)))] \tag{4}$$

where $p_{true}$ is the true data distribution and $p(z|\mu;\sigma)$ is a normal distribution with mean $\mu$ and variance $\sigma$; the random variable z is sampled from it.

### 3.3. Model

In this chapter, we will cover the architectural design and details of the model. The architecture of the model is based on the following considerations. The convective precipitation forecasting is always troubled by the problem of forecasting ambiguity. To address this problem, we try to provide a clearer prediction by setting the overall network architecture as a GAN network. However, GAN networks are known to be vulnerable to pattern collapse, in which the model's predictions fall into several specific patterns and lack diversity. In the specific task of convective precipitation forecasting, the radar echoes predicted by the model tend to have a similar movement trend. It violates the underlying physical constraints. In this study, we believe that this is mainly because the model cannot extract enough spatiotemporal information to identify different motion patterns and guide the model to make accurate predictions. To solve this problem, we try to capture the movement trend of radar echoes of adjacent frames explicitly using a differential calculation. The movement pattern of the radar echo is used as an additional hidden variable to modify the prediction results of the model.

#### 3.3.1. Generator

In our research, a GAN is used to produce a clearer picture of future radar echoes. Compared with the general prediction network, the future radar echo map under a GAN framework can depict a more detailed echo shape, and local weather conditions can also be simulated effectively [34–36]. However, it is well known that GAN networks are prone to fall into the dilemma of gradient disappearance and mode collapse during training. The main reason for the disappearance of the gradient comes from the game process of the generator network and discriminator network. The overlap probability between the distribution of real radar echo data and the distribution of generated radar echo data is very small. Therefore, the discriminator network can easily distinguish the true and predicted distribution during the training process. This makes it difficult for the network to obtain gradient updates to iteratively optimize the network parameters and even makes it difficult to converge during the training process. In addition, mode collapse mainly refers to the possibility that for the data distribution of radar echo prediction results with a multi-modal distribution, the ordinary model tends to predict the probability of a certain fixed radar echo motion mode while ignoring other motion modes. This also results in models that do not fully capture the true data distribution and cover all possibilities for future changes. For the radar echo extrapolation task, we believe that the model cannot extract enough spatiotemporal information to recognize different motion patterns, which is the main reason for the mode collapse. Based on the above reasons, this study uses the differential information between adjacent frames to effectively extract complex high-dimensional spatiotemporal information and simulate capturing the motion trend of the radar echoes.

In the research, the main task of the generator is to use the historical radar echo sequence to generate the radar echo sequence for a certain period of time in the future. At the same time, the landing area, detail information, and overall movement trend need to be consistent with the real radar echo sequence as much as possible. Considering the complex spatiotemporal variability of radar echoes, we hope that the model can extract deeper spatiotemporal characteristic information to produce more accurate predictions. From the perspective of the time dimension, the overall change trend of the atmospheric system will not change so quickly in a short time. Thus, the overall difference between the

two consecutive frames of radar echo images will not be too large. However, local drastic changes are evident, and this important change is often overlooked by previous studies.

Therefore, differences in local weather conditions contain more information about temporal and spatial changes than overall movement trends. By extracting and analyzing the difference information, the model can learn the complex spatiotemporal changes more effectively [28]. In addition, differential technology can explicitly extract the changing part of the adjacent radar echo frame (especially the part with drastic changes), and the invalid background information and unchanging part of the radar echo motion are explicitly ignored. Inspired by this, we use the past frame $X_{t-1}$ and the current frame $X_t$ as differential inputs. In addition, the differential signal is considered to have a strong correlation with the time change. Therefore, we introduce the hidden state $Z_t$, which changes on the timeline, and adopt the gating mechanism to adaptively select the ratio of the difference information and the hidden state at the current time to cope with sudden weather conditions. Finally, we design a spatiotemporal difference LSTM (STD-LSTM) unit, whose structure diagram is shown in Figure 1. Correspondingly, the formula expression of the STR module is shown in Formula (5):

$$
\begin{aligned}
i_t &= \sigma(W_{xi} * \mathcal{X}_t + W_{hi} * \mathcal{H}_{t-1} + b_i) \\
f_t &= \sigma\left(W_{xf} * \mathcal{X}_t + W_{hf} * \mathcal{H}_{t-1} + b_f\right) \\
g_t &= \tanh(W_{xc} * \mathcal{X}_t + W_{hc} * \mathcal{H}_{t-1} + b_c) \\
\mathcal{C}_t &= f_t \circ \mathcal{C}_{t-1} + i_t{}^{\circ} g_t \\
o_t &= \sigma(W_{xo} * \mathcal{X}_t + W_{ho} * \mathcal{H}_{t-1} + b_o) \\
Z_t &= STD(X_t, X_{t-1}, Z_{t-1}) \\
\mathcal{H}_t &= o_t \circ \tanh(\mathcal{C}_t + Z_t)
\end{aligned}
\tag{5}
$$

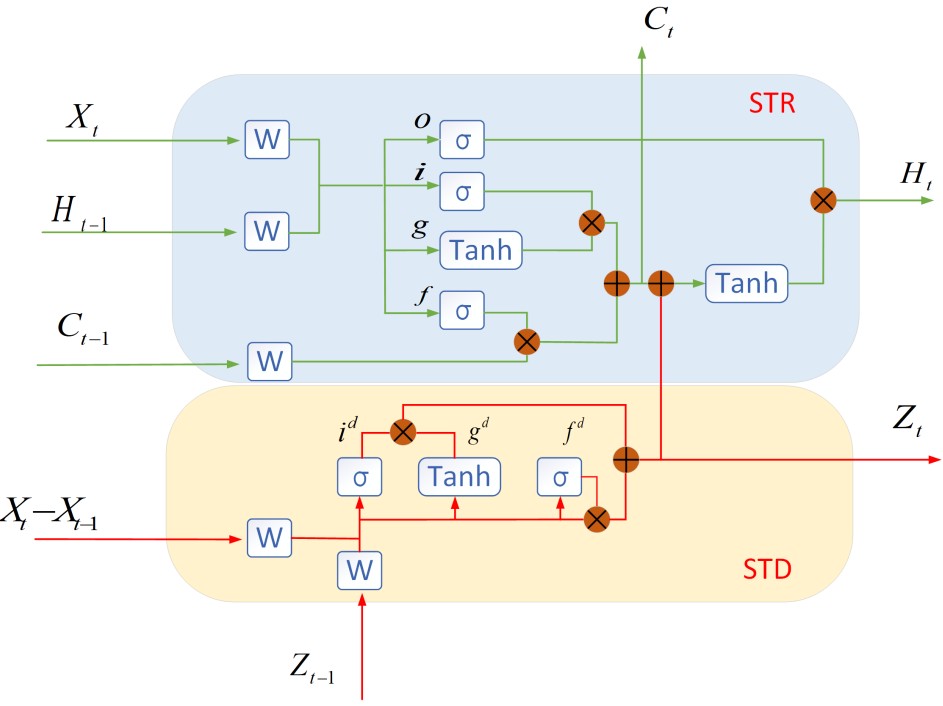

**Figure 1.** The image of the spatiotemporal difference module in our model.

$X_t$ in Figure 1 is the input frame of the current moment, $X_{t-1}$ is the input frame of the previous moment, $H_t, C_t, Z_t$ is the hidden state passed from the unit of the previous moment, and $H_{(t+1)}$, $C_{(t+1)}$, $Z_{(t+1)}$ is the hidden state passed from the difference unit of the next moment. $W$ is a learnable linear or nonlinear transformation. In this study,

$3 \times 3$ convolution kernels are used to achieve state transformation. tanh and $\sigma$ are activation functions. In addition, the detailed calculations of $STD$ are shown as follows:

$$
\begin{aligned}
i_t^d &= \sigma(W_{x'i} * (\mathcal{X}_t - \mathcal{X}_{t-1}) + W_{h'i} * \mathcal{Z}_{t-1} + b_i) \\
f_t^d &= \sigma\left(W_{x'f} * (\mathcal{X}_t - \mathcal{X}_{t-1}) + W_{h'f} * \mathcal{Z}_{t-1} + b_i\right) \\
g_t^d &= \tanh\left(W_{x'g} * (\mathcal{X}_t - \mathcal{X}_{t-1}) + W_{h'g} * \mathcal{Z}_{t-1} + b_i\right) \\
Z_t &= f_t^d \circ Z_{t-1} + i_t^d \circ g_t^d
\end{aligned}
\tag{6}
$$

which takes the memory cells $Z_t$ and the differential features $X_t - X_{t-1}$ as the input. In addition, we use stacked four-layer STD-LSTM cells as our generator network. The specific network structure is shown in Figure 2. It is worth noting that we use the difference between the current input frame and the input frame at the previous moment to capture the spatial difference information. The rich differential information of radar echoes is captured by the layer loop unit to simulate the potential local motion trend. In addition, the hidden state $Z_{t-n:t}$ propagated along the time axis implicitly contains the difference information of radar echoes with long-term changes in the time dimension, which is more helpful for the model to model the movement trend of radar echoes.

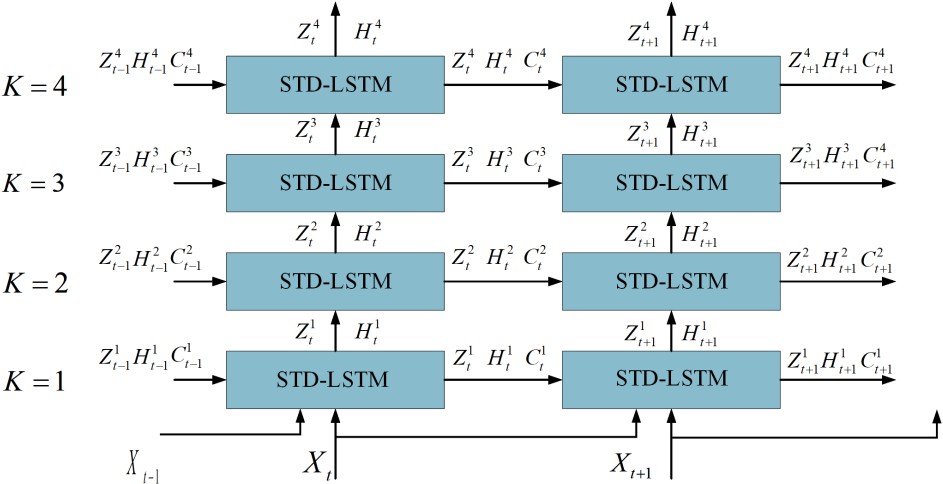

**Figure 2.** Overall image of our model. Our model consists of four layers of STD-LSTM stacked together.

### 3.3.2. Discriminator

The future radar echo maps generated by our generator network will have deeper feature information and richer details for local weather conditions. Therefore, in general, a network with a certain depth is needed to completely extract the feature information of radar echo images and accurately divide the data distribution between the predicted image and the real image. However, deeper networks will inevitably lead to overfitting problems and increased complexity of the training process, which is difficult to apply. Therefore, the inception module [37] is used in this study. In the inception module, convolutional kernels of different sizes are used on the model width to extract radar echo map features of different scale ranges. In addition to feature extraction through convolutional kernels with different receptive fields, radar echo image information is also compressed through downsampling operations and is ultimately converted into a category information output by the discriminator network. Radar echo images contain complex weather conditions, such as the moving track of the large-scale echo and the rotation and dissipation of the small- and medium-scale echoes. Therefore, multi-scale feature lifting in the inception module can effectively extract deep information from both global and local perspectives. In addition, the discriminator can effectively distinguish the real data distribution from

the generated data distribution, which can make the training of the generator easier and the generated radar echo image finer. The detailed design of the discriminator is shown in Figure 3. In experiments, too many module layers will cause training difficulties, and too few module layers will result in an incomplete feature capture. Therefore, we finally adopt a three-layer inception module structure. After the inception module, we use a four-layer feedforward neural network. This can map the extracted spatiotemporal information of radar echoes to a single value as category information for the output of the discriminator.

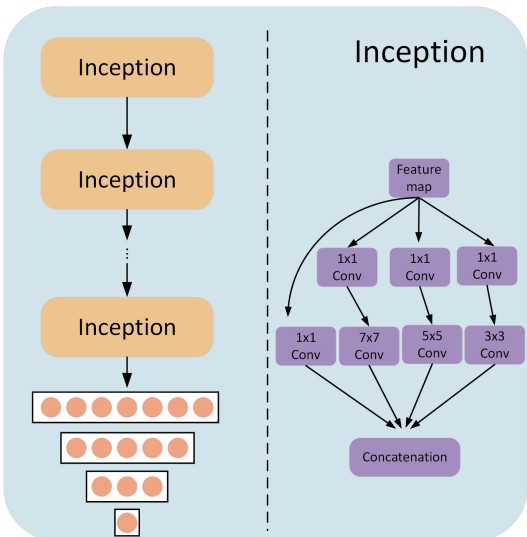

**Figure 3.** Image of the discriminator module. The discriminator consists of three-layer inception modules, finally arranged in a single-channel fashion. Each inception module consists of convolution kernel of $1 \times 1$, $3 \times 3$, $5 \times 5$, and $7 \times 7$.

## 4. Experiment on Radar Dataset

In this chapter, we evaluate the model's performance on a dataset of real radar echoes. The real radar echo datasets include the Shijiazhuang radar echo dataset and the Nanjing radar echo dataset. Our model is mainly divided into a generator network and a discriminator network. Among them, the generator network is formed by stacking cyclic units. We experimented with the model prediction results when the convolution kernel size of the cyclic unit was 1, 3, 5, and 7. An excessive convolution kernel size can affect the computational efficiency of the model and be insensitive to changes in local features. Therefore, we did not experiment with larger convolutional kernel sizes. Finally, based on the experimental results, we set the convolution kernel size of the loop unit to 3. Similarly, we tested the depth of hidden layers in cyclic units at depths of 32, 64, and 128. After weighing the prediction results and memory usage, the model performed best at the hidden layer depth of 64. For the choice of activation function in the recurrent unit, we followed the configuration of the basic RNN model. The discriminator network consists of a three-layer stacked inception module and a fully connected network. The number of hidden channels in the three-layer inception module is 64, 32, and 16, respectively. Our model is optimized using the MSE loss function and BCE loss function and using the ADAM optimizer. In addition, if the learning rate is set too high, it will cause the model to oscillate during the training process or even fail to converge. If the learning rate is set too small, the network will converge very slowly and may even converge into a local extreme point. After testing with the learning rate set in the range of 0.01 to 0.0001, we decided to set the learning rate to 0.001. All experiments were implemented on Pytorch 1.8.

### 4.1. Evaluation Index

In this study, we evaluated the generated radar echoes using the quantitative evaluation metrics MSE (mean square error), SSIM (structural similarity index measure), PSNR (peak

signal-to-noise ratio), CC (correlation coefficient), and CSI (critical success index). The MSE measures the difference between the real data and predicted data, and the smaller the MSE, the better the quality of the data generated. The SSIM ranges from −1 to 1, with higher scores indicating better similarity between the real data frame and the generated results. PSNR scores are positive, and higher grades indicate better performance. The CC measures the correlation between the real radar echoes and the predicted radar echoes. The higher the value of the CC, the closer it is to the true value. We used the rainfall intensity threshold of 0.5 mm/h to calculate the CSI [38]. We first converted the pixel value of prediction or groundtruth images to 0 or 1 by threshold $\tau$ mm/h. In detail, we used the Z-R relationship [39] to convert the pixel values to rainfall R. If R >= $\tau$, the pixel value will be 1; otherwise, the pixel value will be 0. Then, we can calculate the true positive (TP) (prediction = 1, truth = 1), false negative (FN) (prediction = 0, truth = 1), false positive (FP) (prediction = 1, truth = 0), and true negative (TN) (prediction = 0, truth = 0) separately. In the end, the CSI score is calculated as (TP/(TP + FN + FP)).

### 4.2. Shijiazhuang Radar Echo Dataset

In this study, we used the radar echo dataset with raw radar data at polar coordinates as the data source. Among them, the radar echo data collection is the dual-polarization radar data with a time resolution of 6 min provided by the Shijiazhuang Meteorological Bureau of Hebei Province during 2019–2022. Predicting the track and shape change of radar echoes plays an important role in guiding flood control and disaster relief and in reducing the negative impact on human social activities. The complex variations in the shape, accumulation, and dissipation of radar echoes mean that this is a challenging task. The following details are our data processing operations.

Radar data are usually stored in binary format in polar coordinates. However, the calculation of the neural network does not accept data in the polar coordinate format, so we needed to rasterize the radar data. We used K-nearest neighbor interpolation to rasterize the data into a 200 × 200 grid, which covered the entire area of Shijiazhuang City (113.5°E–115.5°E and 37°N–39°N with a spatial resolution of 1 km). Considering that the radar is susceptible to interference from terrain and other factors in the scanning process, we took the following steps to control the quality of the radar echoes. First, we suppressed the influence of ground clutter and isolated echoes as much as possible, and we used the combined reflectivity factor (i.e., the maximum radar echo reflectance value at each azimuth and distance interval) as the pixel value of our rasterized radar echo data.

#### 4.2.1. Implementation

We mapped the radar signal strength to a range of zero to one and used bilinear interpolation to shape it to a size of 200 × 200. Weather radar data are generated every six minutes, so there are 240 frames per day. To obtain a disjoint dataset for training and testing, we divided each daily sequence into six blocks and randomly assigned five blocks for training and one block for testing. We then sliced successive frames in each block with a 20-frame wide sliding window. Therefore, the total sequence was divided into a training set of 4000 samples and a test set of 400 samples. We first referred to the general setup for performing a spatiotemporal series prediction task on a radar dataset by generating 10 future frames given 10 previous frames while training the model. We then extended the extrapolation length from 10 segments to 20 segments to explore the ability of the compared models to cover long-term predictions for the next 2 h. After the prediction, we show the dBZ intensity in Figure 4.

#### 4.2.2. Result

As can be seen from the prediction of the radar returns over the next two hours in a period of severe convective weather provided in Figure 4, our model can provide a finer and more accurate picture of qualitative results. The ConvLSTM was able to estimate the overall position of the radar echoes only in the first 10 frames of the prediction, but the

prediction results in the last 10 frames deviated from the trajectory of the real radar echoes. MIM and PredRNN++ could provide relatively accurate echo intensity compared with ConvLSTM, but the overall and local echo shape and motion trajectory were still not accurately captured. This suggests that past models have struggled to predict complex weather changes. In addition, although MotionRNN could accurately predict the overall location of large-scale clouds, it could not describe the details of local weather conditions. In contrast, our spatiotemporal differential module could capture differential signals in the atmospheric environment to simulate the movement trend of cloud clusters at various scales as reflected by radar echoes. Specifically, it could accurately capture the movement of large-scale clouds and effectively predict the disappearance and appearance of small-scale clouds.

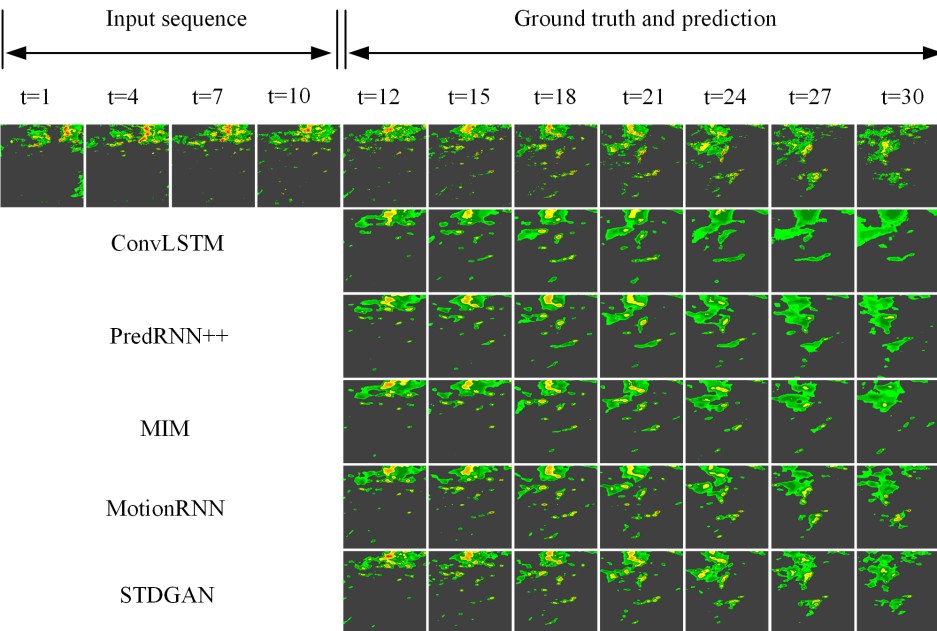

**Figure 4.** Visualizationsamples on the Shijiazhuang radar echo dataset. All models used the first ten frames as the input and predict the last twenty frames. The temporal resolution of the dataset is six minutes, and this prediction task tests the performance of the models in a long-term and complex prediction environment.

On the other hand, the accuracy of the other comparison models rapidly declined over time, while ours could maintain a slow rate of decline. In addition, from the quantitative results of all data in the test set provided in Table 1, it can be seen that our model could maintain optimal results on the four evaluation indicators (MSE, PSNR, SSIM, and CSI).

**Table 1.** Quantitative results of our model and advanced models on the Shijiazhuang radar echo datasets. We compared the MSE, PSNR, SSIM, and CSI of all models on the test set. The lower the MSE and the higher the PSNR, SSIM, and CSI, the higher the prediction accuracy.

| Method | MSE | PSNR | SSIM | CSI |
|---|---|---|---|---|
| ConvLSTM (NIPS 2015) [1] | 209.45 | 24.78 | 0.725 | 0.577 |
| PredRNN++ (PMLR 2018) [27] | 169.10 | 25.23 | 0.760 | 0.589 |
| MIM (CVPR 2019) [28] | 159.94 | 24.83 | 0.767 | 0.605 |
| MotionRNN (CVPR 2020) [29] | 145.05 | 36.18 | 0.775 | 0.608 |
| STDGAN | 132.22 | 37.87 | 0.796 | 0.612 |

However, Table 1 only shows the overall performance of the model over a test time scale. The performance of the model in the minimum time resolution of radar echoes is also of great reference value. Therefore, we conducted statistics on the performance of

each time node of the model in all test sets at the 6-minute scale. As can be seen from Figure 5, under the four evaluation indexes of MSE, CC, SSIM, and PSNR, the prediction performance of our model was still far better than that of other models for every 6 min in the next 2 h. Moreover, the performance decline trend of our model was much slower than that of the other models, indicating that our model has good stability. Specifically, the slow-growing MSE indicates that the predicted results of our model differed the least from the real future radar echoes. Due to the complexity of weather conditions, a complete weather process includes many local meteorological changes. Therefore, the lower MSE also indicates that the forecast results of our model are more effective at simulating small local weather conditions. In addition, the slow decline rates of the SSIM, PSNR, and CC indicate that our image quality is stable at a certain level, which has reference value for the business work of forecasters. As a result, our model can assist business weather forecasting tasks while providing good prediction results for complex cloud formation and disappearance phenomena.

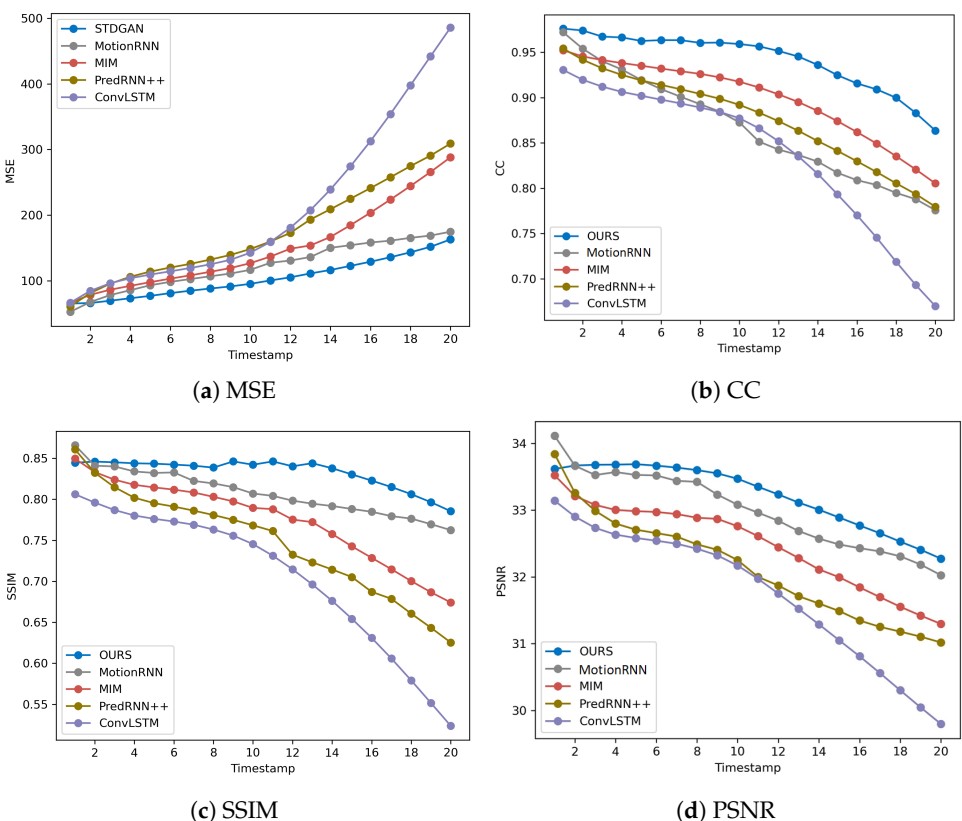

**Figure 5.** Frame-wise comparisons of the next 20 extrapolation echoes on the Shijiazhuang Radar Echo Dataset.

### 4.3. Nanjing Radar Echo Dataset

In addition to the radar echo data provided by the Shijiazhuang Meteorological Bureau, we also used the publicly available dual-polarized radar echo dataset provided by Nanjing University to verify the prediction capability of the model. The publicly available dataset allows readers to easily replicate and validate the prediction effect of our model. The dataset contains 268 precipitation events collected by Nanjing University between 2014 and 2019 at a spatial resolution of 1 km and a temporal resolution of 6–7 min, and it is available at https://doi.org/10.5281/zenodo.5109403 (accessed on 13 October 2021). Unlike the Shijiazhuang radar echo dataset, the study area of this expedition is a $256 \times 256$ km area centered on the radar. In addition, the dataset has been processed for quality control [40]. Therefore, we did not perform redundant preprocessing in our experiments. Regarding this

radar echo dataset, we present the quantitative and qualitative comparisons of the models in convective and stable precipitation scenarios.

### 4.3.1. Implementation

We have divided the dataset into a training set containing 6000 sequences and a test set containing 500 sequences. Each sequence in the training set is 20 frames in length, and the sequences in the test set are 30 frames in length. The first ten frames are observation frames and the last ten and twenty frames are real frames for comparing the model predictions.

### 4.3.2. Result

Take the case of convective rainfall. Figure 6 provides the plots of the convective rainfall results of our model and the comparison models for predicting the next two hours on the test set. Convective precipitation is usually localized, intense, and unevenly distributed. This makes it highly challenging for the prediction model to handle local and high-intensity rainfall fields. Specifically, as shown in Figure 6, our model was able to provide richer and closer echo patterns to the original radar echoes compared with the other models. In addition to the overall echo shape, the prediction efficiency of STDGAN for small echoes was also better than the other compared models. Over time, the compared models gradually lost the correct predicted trajectories, while STDGAN retained the echo information better. This suggests that the way STDGAN fuses the cyclic unit and the difference mechanism can better capture local weather patterns. In addition, this mechanism also has better fitting ability for small echoes that can be easily ignored.

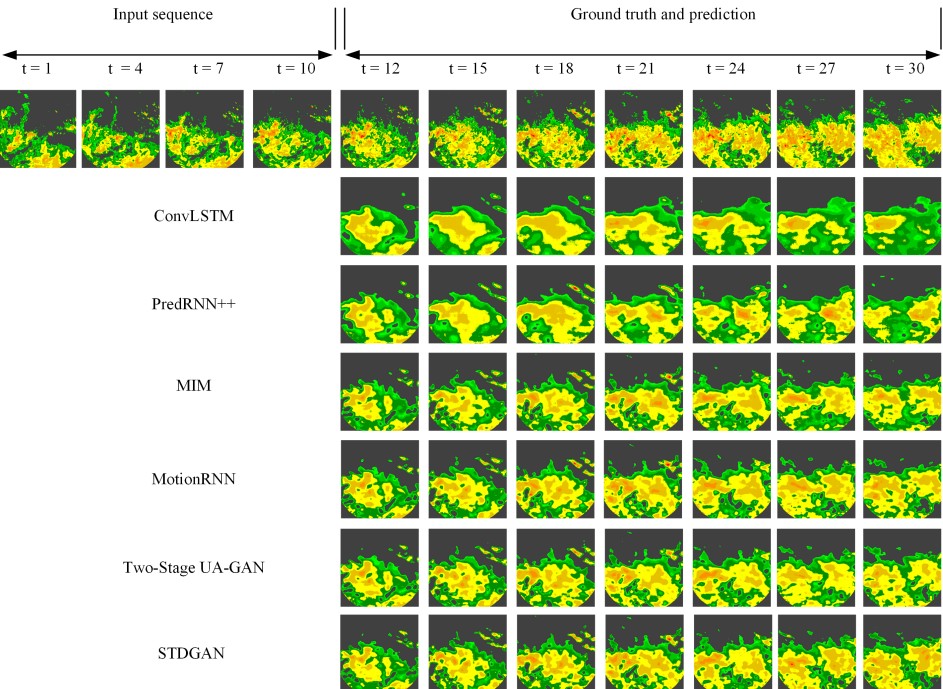

**Figure 6.** Sample visualization of model prediction results in the convective precipitation scenario. In the Nanjing radar echo data prediction mission, all models use the first ten frames as the input and predict the next twenty frames. The temporal resolution of the dataset is six minutes.

Compared with convective precipitation, stable precipitation is usually continuous, with relatively small precipitation intensity but wide coverage. Figure 7 shows a qualitative comparison between STDGAN and the comparison models under stable precipitation scenarios. ConvLSTM and PredRNN can only predict the rough coverage range of radar echoes over a large range. MIM, MotionRNN, and Two-Stage UA-GAN are less sensitive at predicting local radar echo intensity. STDGAN predicts the overall and local echo states more accurately and comprehensively in stable precipitation fields.

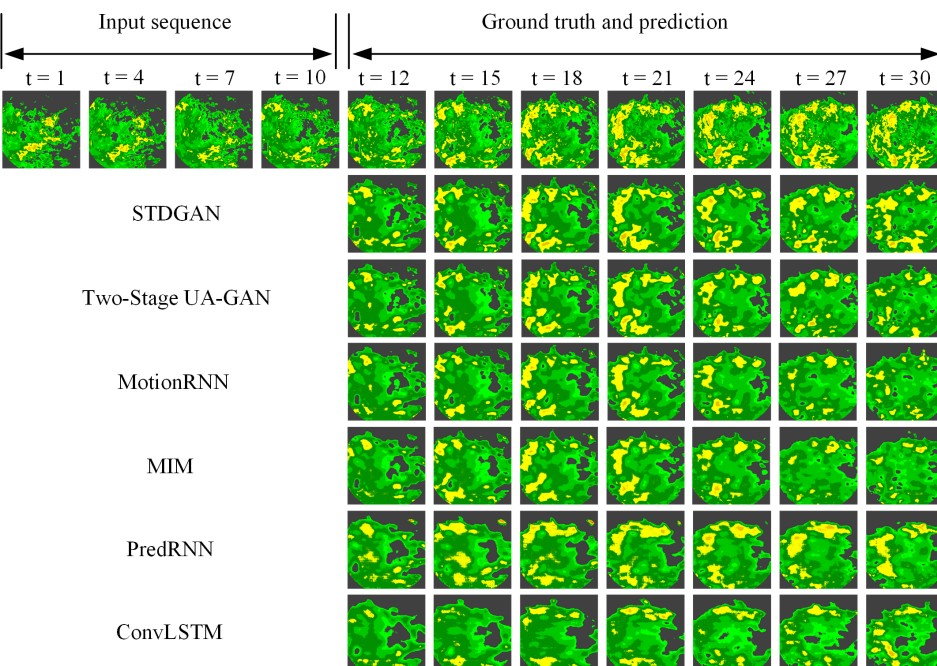

**Figure 7.** Sample visualization of model prediction results on the Nanjing radar echo dataset. As in the Shijiazhuang radar echo data prediction task, all models use the first ten frames as the input and predict the next twenty frames. The temporal resolution of the dataset is six minutes.

Table 2 numerically compares the prediction results of our model and the comparison models using five evaluation metrics: MSE, SSIM, PSNR, CC, and CSI. We selected the time nodes of the next 20 frames for comparison. At all time nodes, STDGAN outperformed the previous models in terms of accuracy in radar echo prediction. In addition, STDGAN could provide better image evaluation indicators at the radar echo image level. Through the CSI indicators, STDGAN could hit more rainfall grid points in scenarios with a rainfall threshold of 0.5 mm/h compared with the other comparative models.

**Table 2.** Quantitative evaluation of different methods in the open radar echo dataset of Nanjing University under convective precipitation scenarios. These metrics are averaged over 20 predicted frames. Lower MSE values are better, and higher PSNR, SSIM, CC, and CSI values are better. Bold font represents the optimal result for each indicator.

| Method | MSE | PSNR | SSIM | CC | CSI |
|---|---|---|---|---|---|
| ConvLSTM [1] | 62.412 | 0.504 | 28.414 | 0.653 | 0.493 |
| PredRNN++ [27] | 52.421 | 0.70 | 29.951 | 0.753 | 0.502 |
| MIM [28] | 52.224 | 0.698 | 29.853 | 0.718 | 0.510 |
| MotionRNN [29] | 51.626 | 0.698 | 29.861 | 0.744 | 0.512 |
| Two-Stage UA-GAN [41] | 50.796 | 0.701 | 29.965 | 0.755 | 0.518 |
| STDGAN | **49.570** | **0.714** | **30.633** | **0.855** | **0.523** |

Similarly, we used the prediction results of the model at each frame to compare its extrapolation and information extraction capabilities at each time node. Figure 8 shows the MSE, PSNR, SSIM, and CC for each of the predicted 20 frames for all tests. Based on these indicators, we can come to the conclusion that our model has higher image clarity, lower variance loss, and higher image quality in most cases. Over time, the other models showed a significant decrease in predictive performance, while our model showed less decline than the other models.

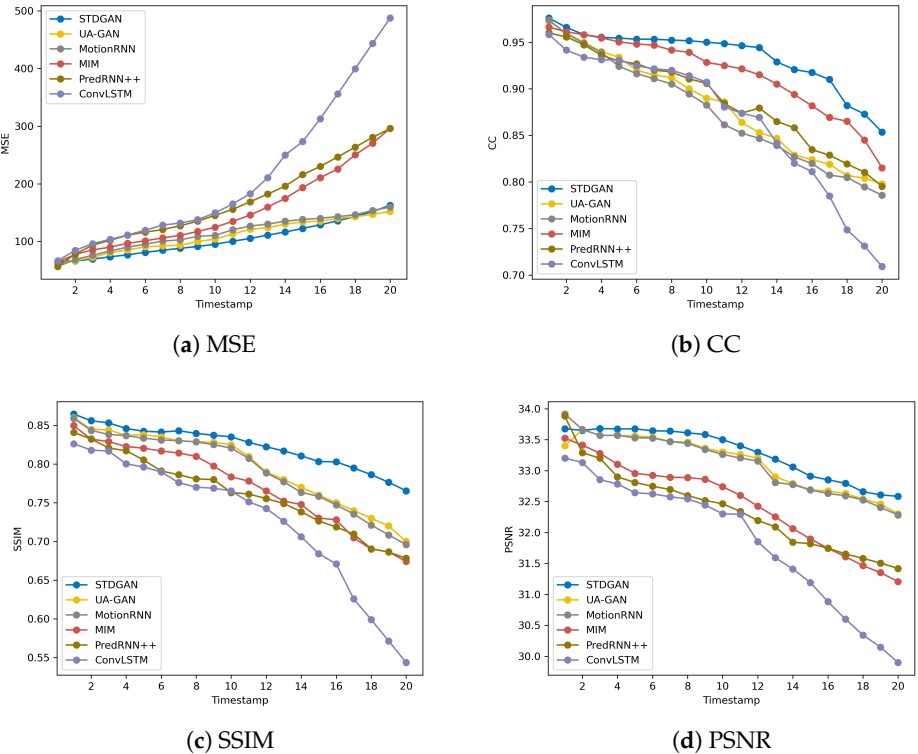

**Figure 8.** Frame-wise comparisons of the next 20 extrapolation echoes on the Nanjing radar echo dataset.

### 4.4. Ablation Experiment

Given the modular composition of our model, ablation experiments were conducted to analyze the contribution of each module to the final predicted results. Table 3 shows the quantitative results of our model in full modules, without the spatiotemporal difference module (STD), and without the discriminator module, respectively. As can be seen from Table 3, STDGAN under the complete module has a strong ability to fit the movement changes of radar echoes, and its MSE score exceeds that of most previous models based on an RNN structure. This indicates that the difference between the image predicted by STDGAN and the ground truth value of the corresponding pixel is small. In addition, the approach based on SSIM and PSNR makes it closer to the original image in brightness, contrast, and structure. When the STD module is removed from the model, the indexes of all aspects of the model decrease significantly. We believe that this is because the model cannot explicitly learn the time difference of adjacent frames, which leads to the incomplete extraction of the spatiotemporal information of radar echoes. After removing the discriminator, the evaluation indexes decreased compared with the complete model. Because the discriminator restricts the distance between the real data distribution and the generated data distribution, it can provide a more realistic and detailed radar echo image. Therefore, when the discriminator is removed, the model often encounters the multi-peak Gaussian data distribution, and the prediction result is fuzzy due to the uncertainty distribution.

**Table 3.** Model ablation experiments on the Shijiazhuang radar echo dataset. STD represents the spatiotemporal difference module.

| Method | MSE | PSNR | SSIM |
|---|---|---|---|
| STDGAN | 132.2 | 37.87 | 0.796 |
| STDGAN (W/O STD) | 139.47 | 37.04 | 0.786 |
| STDGAN (W/O Discriminator) | 137.49 | 36.93 | 0.742 |

## 5. Discussion

This study used real radar echo data provided by the Shijiazhuang Meteorological Bureau and Nanjing University. Among these data, the raw radar echo base data provided by the Shijiazhuang Meteorological Bureau contain dirty data and missing values, which will affect the training effectiveness of the model. The dirty data are mainly due to clutter interference during radar observation, which interferes with the data acquisition process. Therefore, K-nearest neighbor interpolation and quality control were used to solve the problems on the Shijiazhuang radar echo dataset. The Nanjing radar echo dataset has undergone quality control. Therefore, we did not perform any additional preprocessing. Finally, we obtained radar echo datasets from the Shijiazhuang and Nanjing datasets at time resolutions of 6 min and spatial resolutions of 1 km, respectively. In the experiment, we evaluated the predictive performance of STDGAN and previous models on the dataset using both qualitative and quantitative indicators. In terms of quantitative results, STDGAN had a lower MSE and higher SSIM, PSNR, CC, and CSI compared with the previous models studied. In terms of qualitative results, ConvLSTM could only predict fuzzy and incomplete radar echo images. PredRNN++ and MIM had weak predictive ability for strong echoes and could only predict the overall contour of the echoes. MotionRNN's prediction of radar echo intensity was superior to the previous models. However, the detailed prediction of local echoes was still not accurate enough. The Two-Stage UA-GAN also adopts a GAN structure and focuses on local weather patterns. However, STDGAN achieved a higher efficiency and could obtain more comprehensive local information by performing differencing between adjacent frames. Therefore, STDGAN's prediction images of local meteorological patterns and overall radar echoes are closer to real radar echo images in terms of prediction clarity and accuracy. In addition, STDGAN accumulates fewer prediction errors, indicating its advantage in long-term predictions.

## 6. Conclusions

We propose a model for radar echo extrapolation, STDGAN. Most of the existing models only extract the temporal and spatial features roughly but ignore the great value of local detail features for rainfall prediction. The innovation of our model is that each cycle unit adds a spatiotemporal difference module. This allows the model to better capture and model the movement of local, small-scale weather patterns, thus providing more accurate predictions of the overall radar echo change pattern. In addition, our model performs spatiotemporal series prediction in the framework of GAN networks. The multi-scale feature extraction of the discriminator can effectively model the feature information of weather models of various scales. Finally, the adversarial mode inherent in GAN networks can make the distribution of generated data closer to the distribution of real data, thus providing a clearer radar echo image. The effectiveness of the model was verified using experiments on radar echo datasets. The experiments on radar echo datasets in Shijiazhuang and Nanjing have shown that STDGAN has advantages over previous models in predicting local meteorological patterns and overall echo movement trends. And, in long-term predictions, STDGAN can maintain a better prediction performance.

**Author Contributions:** Conceptualization, C.W., W.T. and K.S.; methodology, C.W., W.T. and K.S.; validation, C.W., W.T. and K.S.; writing—original draft preparation, C.W., W.T. and K.S.; writing—review and editing, C.W., W.T. and K.S.; visualization, C.W., W.T. and K.S.; supervision, X.N., L.Z. and B.L.; project administration, X.N., L.Z. and B.L.; funding acquisition, X.N., L.Z. and B.L. All authors have read and agreed to the published version of the manuscript.

**Funding:** This work was funded by State Key Laboratory of Geo-Information Engineering NO.SKLGIE2020-M-4-2, in part by the National Key Research and Development Program of China under Grant 2021YFE0116900, in part by the National Natural Science Foundation of China under Grant 42075138, and in part by the National Natural Science Foundation of China under Grant 42175157.

**Data Availability Statement:** All data included in this study are available upon request by contacting the corresponding authors.

**Conflicts of Interest:** The authors declare no conflict of interest.

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
