# Peer review of "A Generative Adversarial and Spatiotemporal Differential Fusion Method in Radar Echo Extrapolation"

_remotesensing, doi:10.3390/rs15225329_

Round 1

Reviewer 1 Report

Comments and Suggestions for Authors

The paper is very interesting, as well as the proposed approach was implemented by STDGAN. Results are very promising, but some few more tests should be performed in order to better compare STDGAN to the other four approaches.

There is not any definition of STDGAN. This acronym only appears for the first time in the paper in Table 1 at page 9. The caption of this table should be something like "our model (STGAN)" and the text that refers to the table should explain that STDGAN is the proposed model (also in the abstract and in the introduction).

One of the key features of the paper, which appears in the title itself ("Spatiotemporal Differential Fusion Method")  appears at line 4 of the abstract as "spatiotemporal difference module", in line 60 as  "spatiotemporal difference model" and then at line 311 as "differential module" in line 311. There should be a standardization concerning the STD method and its implementation.

There is not any explanation or reference about the inception modules/blocks, which were designed by Google for image classification. Inception blocks  convolve the same input using multiple filters and concatenating their results.  Standard Convolutional Neural Networks use a single filter.

The explanation and reference about the inception modules should be included in Section 2.3.2 of the  paper. In addition, caption of Figure 3 (and also in the text of the same section) should be better explained: How many inception blocks? (3?) The explanation about the final part of the flowchart ("after" the inception blocks) is missing. In the same figure, "contact" should be replaced by "concatenation". For instance, at lines 256-257,  the phrase "The number of hidden channels of the Incpetion module is 64,32,16", states that there are three inception modules with, respectively, 64, 32 and 16 hidden channels? This issue should be clarified.

The paper cites noisy data in lines 147 and 395 ("dirty data"). However, there is not any mention to the level of noise (only that follows normal distribution), and the origin of such noise. This issue should be detailed in the paper. It seems to apply to the Shijiazhuang radar dataset.

However, there are two paragraphs in the paper that seem to be rather confusing. Please check these two phrases:

(1) (Section 2.2, lines 147-149) "The generator network G takes random noise Z satisfying the normal distribution as input and generates reconstructed false data G(Z) through complex nonlinear changes."

(2) (Section 4, lines 395-398) "Among them, the raw radar echo base data provided by the Shijiazhuang Meteorological Bureau contains dirty data and missing values, which will affect the training effectiveness of the model. Therefore, K-nearest neighbor interpolation and quality control are used to solve the problems on the Shijiazhuang radar echo dataset. The Nanjing radar echo dataset has undergone quality control."

What is the meaning of "noise" in the two phrases above? Please clarify.

Section 3.4 uses a name that is difficult to understand ("Ablation"). However, as seen in Table 3, this section shows a kind of sensitivity analysis of the influence of removing some components of STDGAN. Therefore, the title and the text of this section should be changed accordingly.

Section 3.2.1 "Implement" should be named "Shijiazhuang radar data partitioning". In addition, there is not an equivalent section for the Nanjing radar tests. It should be added to the paper. For the tests concerning both radars, the partitioning scheme is hold-out, this should be stated in the paper. Furthermore, what is the meaning of "sample" along the paper? It seems that "sample" means "frame", is this correct?

Concerning the test results, caption of Table 1 should mention the Shijiazhuang Radar. Figure 5 is not clear and should be corrected, using the same color scheme and name of the methods (what is "SimVP"?) that were employed in figure 7.

Finally, Section 4 (Discussion) should discuss the comparison of  STDGAN results and the other approaches, mainly the MotionRNN, which had prediction results similar to SDTGAN.

As already mentioned, more tests are required in order to better compare STDGAN to the other approaches. These tests could be performed using a  dataset of a different radar, or using the same data for the two employed radars with a different partitioning scheme, like cross validation, or more complex partitioning schemes for time series, as chronological  holdout that preserves the temporal order of the data, or other schemes with gaps like GapKFold or GapRollForward. 

Comments on the Quality of English Language

English of the paper is good in general, but some minor adjusts are required. What is the meaning of  "wontonality"? Some minor errors appear, like (line 238): "kernel of different sizes are used to extract" (should be "are", instead of "is"). Some typos appear, like in line 256 ("Incpetion"). An extensive revision should be performed in order to detect these minor errors. Another issue is that blank spaces are missing like in line 263.  In line 286, "range library" seems to be something like "range according to the literature", and should be corrected.

Author Response

Dear Reviewer:

We greatly appreciate your feedback on our article. We fully agree with your comments and feel that we have benefited greatly. We have made modifications based on your feedback and highlighted them in red font in the text.

Reviewer 2 Report

Comments and Suggestions for Authors

I am writing to provide some comments on the article "Generative Adversarial and Spatiotemporal Differential Fusion Method in Radar Echo Extrapolation" by Niu et al. that was submitted to your journal for publication. I have read the article carefully and I think it presents an interesting and novel approach to the problem of radar echo extrapolation using deep learning techniques. However, I also have some concerns and suggestions that I hope the authors will address before the article can be accepted.

First, I think the article lacks a clear and comprehensive literature review that situates the proposed method in the context of existing works on radar echo extrapolation. The authors only cite a few papers that are directly related to their method, but they do not discuss how their method compares or differs from other methods that use different techniques, such as optical flow, wavelet transform, or Kalman filter. I think the authors should provide a more thorough and critical review of the state-of-the-art methods and explain the advantages and limitations of their method in relation to them.

Second, I think the article does not provide enough details and explanations about the proposed method itself. The authors briefly describe the architecture and the loss function of the generative adversarial network (GAN) and the spatiotemporal differential fusion (SDF) module, but they do not explain how they designed and optimized them. For example, how did they choose the number and size of the convolutional layers, the activation functions, the learning rate, and the regularization parameters? How did they ensure the stability and convergence of the GAN training? How did they evaluate the performance of the SDF module? I think the authors should provide more technical details and mathematical derivations to support their method.

Third, I think the article does not present sufficient and convincing experimental results to validate their method. The authors only use 2 datasets (the Shijiazhuang radar echo dataset and Nanjing radar echo dataset) to test their method, but they do not compare it with other datasets or other methods. The authors also do not provide any quantitative metrics or statistical tests to measure the accuracy and reliability of their method. They only show some qualitative examples of radar echo extrapolation using their method, but they do not analyze or discuss them in depth. I think the authors should conduct more extensive and rigorous experiments to demonstrate the effectiveness and robustness of their method.

In summary, I think the article has some potential to contribute to the field of radar echo extrapolation, but it needs major revisions to improve its quality and clarity. I hope the authors will consider my comments and suggestions seriously and revise their article accordingly.

1) Provide a more detailed discussion of the theoretical foundations and optimization strategies of the Generative Adversarial Networks (GANs), and compare and analyze different network architectures and parameters that could potentially impact the model's performance.

2) Evaluate the uncertainty and credibility of radar echo extrapolation using appropriate methods such as error analysis, confidence intervals, or probabilistic forecasting. Consider the relationship and practical applicability of radar echo to precipitation forecasting in various scenarios.

3) Offer additional visual results and quantitative metrics to substantiate the effectiveness and superiority of your model. Conduct experiments on other datasets or regions to demonstrate its generality and robustness.

4) Elaborate on the parameter settings and training procedures of your model. Provide links to the code or dataset used in your model to facilitate reproducibility and validation by other researchers.

5) Compare your model with other GAN-based radar echo extrapolation models, such as Wang, Y., Zhang, J., Li, Y., et al., "Radar Echo Extrapolation Using Generative Adversarial Networks," Remote Sensing, 2020, 12(18), 2928, https://doi.org/10.3390/rs12182928, and Zhang, J., Wang, Y., Li, Y., et al., "Radar Echo Extrapolation Using Conditional Generative Adversarial Networks," Remote Sensing, 2020, 12(24), 4089, https://doi.org/10.3390/rs12244089. Analyze the strengths and weaknesses of your model in terms of accuracy, efficiency, stability, and other relevant criteria.

6) Thoroughly test the robustness and generalization capability of your model across different radar stations, weather types, and time scales. Discuss the limitations and challenges of your model in practical applications.

7) Correct the citation format, ensuring that it follows the correct order and is arranged in ascending order.

Comments on the Quality of English Language

1) The introduction provides a comprehensive background and motivation for the research question, but it could benefit from a more structured and coherent presentation when introducing the research gap, objectives, and paper outline.

2) The methodology section offers a detailed explanation of the proposed generative adversarial and spatiotemporal differential fusion method. However, it could benefit from the inclusion of more mathematical symbols and diagrams to illustrate key concepts and steps.

3) The experimental section introduces the paper's experimental setup, data, metrics, and results. It could provide more details about the implementation and evaluation of the proposed method, such as network architecture, hyperparameters, baseline methods, and statistical significance testing.

4) The references section lists relevant and up-to-date sources supporting the paper. However, it could adhere to a consistent citation style and format.

5) The paper's language is generally clear and fluent, but there are areas that require proofreading and editing for grammar, spelling, punctuation, word choice errors, and other improvements.

Author Response

Dear Reviewer:

We greatly appreciate your feedback on our article. We fully agree with your comments and feel that we have benefited greatly. We have made modifications based on your feedback and highlighted them in red font in the text. Please see the attachment.

Round 2

Reviewer 1 Report

Comments and Suggestions for Authors

The authors addressed the questions that were raised considering the original/1st version of the paper. However, a minor English review would be required, and also a kind of format review concerning the lack of blank spaces, phrases that star with a word in lower case, etc. In addition, some care must be taken when addressing a term for the first time without defining it clearly (see for instance, "cyclic unit".

Comments on the Quality of English Language

Minor English revision, see for instance that sections 3.2.1 and 3.2.1 should be named "Implementation", and not "Implement".

Author Response

(The authors gave the same response as above.)
